

# Systematic evaluation of the gut microbiome of swamp eel (*Monopterus albus*) by 16S rRNA gene sequencing

Xuan Chen[1,*], Shaoming Fang[2,*], Lili Wei[3] and Qiwang Zhong[1]

[1] Jiangxi Engineering Laboratory for the Development and Utilization of Agricultural Microbial Resources, College of Biological Science and Engineering, Jiangxi Agricultural University, Nanchang, China
[2] College of Animal Science, Fujian Agriculture and Forestry University, Fuzhou, China
[3] College of Animal Science and Technology, Jiangxi Agricultural University, Nanchang, China
* These authors contributed equally to this work.

Corresponding authors
Qiwang Zhong,
zhongqw2000@163.com
Lili Wei, hbliliwei@163.com

## ABSTRACT

**Background:** The swamp eel (*Monopterus albus*) is a commercially important farmed species in China. The dysbiosis and homeostasis of gut microbiota has been suggested to be associated with the swamp eel's disease pathogenesis and food digestion. Although the contributions of gut microbiome in fish growth and health has been increasingly recognized, little is known about the microbial community in the intestine of the swamp eel (*Monopterus albus*).
**Methods:** The intestinal microbiomes of the five distinct gut sections (midgut content and mucosa, hindgut content and mucosa, and stools) of swamp eel were compared using Illumina MiSeq sequencing of the bacterial 16S rRNA gene sequence and statistical analysis.
**Results:** The results showed that the number of observed OTUs in the intestine decreased proximally to distally. Principal coordinate analysis revealed significant separations among samples from different gut sections. There were 54 core OTUs shared by all gut sections and 36 of these core OTUs varied significantly in their abundances. Additionally, we discovered 66 section-specific enriched KEGG pathways. These section-specific enriched microbial taxa (e.g., *Bacillus*, *Lactobacillus*) and potential function capacities (e.g., amino acid metabolism, carbohydrate metabolism) might play vital roles in nutrient metabolism, immune modulation and host-microbe interactions of the swamp eel.
**Conclusions:** Our results showed that microbial diversity, composition and function capacity varied substantially across different gut sections. The gut section-specific enriched core microbial taxa and function capacities may perform important roles in swamp eel's nutrient metabolism, immune modulation, and host-microbe interactions. This study should provide insights into the gut microbiome of the swamp eel.

## INTRODUCTION

The swamp eel (*Monopterus albus*), taxonomically belonging to order Synbranchiformes, family Synbranchidae, is an air-breathing teleost widely distributed in swamps, streams, ponds, and ricefields of southern China, Japan, India and other Southeast Asian countries (FishBase, http://fishbase.org/). Due to its great growth performance and rich nutrient content, swamp eel has become a commercially important farmed species in China (*Li et al., 2017*) with the production of swamp eel reaching 358,295 tons in 2017. Diseases and low feed efficiency are two major factors restricting the development of swamp eel aquaculture. Probiotics are live microorganisms that confer a health benefit on the host when administered in adequate amounts (*De et al., 2014*). Many studies have revealed that probiotics can modulate gut microbial balance, enhance immune status, reduce disease susceptibility, and improve feed efficiency (*Caruffo et al., 2016*; *Hai, 2015*; *Newaj-Fyzul & Austin, 2015*). It has been suggested that the dysbiosis and homeostasis of gut microbiota might be associated with swamp eel's disease pathogenesis and food digestion.

Earlier research on the gut microbiota of freshwater and marine fishes has demonstrated that gut microbiota played a crucial role in host nutrient metabolism, growth and health. Many cellulose-decomposing bacteria were shown to be harbored in the intestine of grass carp (*Ctenopharyngodon idellus*), such as *Anoxybacillus*, *Actinomyces*, and *Citrobacter* (*Wu et al., 2012*). When fed with commercial pellet, Pompano (*Trachinotus blochii*) has shown a high abundance of *Clostridia* which is associated with polysaccharide degradation (*Rasheeda et al., 2017*). Alpha diversity and dominant bacterial taxa significantly changed with the development of *Siniperca chuatsi* (*Yan et al., 2016*). The interactions between threespine stickleback (*Gasterosteus aculeatus*) and gut microbiota played a key role in the development of their gut innate immunity (*Small et al., 2017*). Moreover, gut microbial communities in different gut sections exhibited distinct differences in diversity and richness. The alpha-diversity indices in the midgut (called foregut in *Ye et al., 2014*) were significantly higher than in the hindgut in both Asian silver carp and gizzard shad. In salmon, microbial richness was higher in the digesta than in the mucosa (*Gajardo et al., 2016*); however, in the rabbitfish (*Siganus fuscescens*), microbial richness significantly increased from content section to mucosal section (*Nielsen et al., 2017*). Since gut microbiome are complex and dynamic communities with a profound influence on fishes, it is important to systematically characterize the bacterial communities in different gut sections. To the best of our knowledge, there have been few studies on the gut microbiome of the swamp eel.

*Monopterus albus* is a strict carnivore that preys on fishes, worms, crustaceans and other small aquatic animals in the wild (*Liem, 1967*). Under captive conditions, swamp eel are usually fed with surimi or a mixture of commercial power feed. The gastrointestinal tract of the swamp eel is a straight, uncoiled tube passing directly to the anus, and includes the pharynx, esophagus, stomach, midgut (ileum) and hindgut (rectum). Bile enters the midgut by way of a short ductus choledochus (*Liem, 1967*). There is no external demarcation between mid- and hindgut and histologically, the midgut and the hindgut are also similar in that both contain four tunics: mucosa, submucosa, muscularis, and serosa.

However, there are some microscopic differences. First, the mucosal folds of the midgut are in a reticular configuration, while the mucosal folds of the hindgut are distinctly longitudinal and not as numerous as in the midgut. Second, in the midgut, the mucus secreting goblet cells are extremely numerous, while in the hindgut, the number of goblet cells is a food dependent feature, and starvation can cause a pronounced decrease in rectal goblet cells. Third, the serosa of the hindgut is much more prominent than that of the midgut. These make swamp eel likely to contain specific intestinal microbiota similar to those of carnivorous fish, such as the phylum *Cetobacterium, Clostridium,* or *Fusobacteria.* The microscopic differences in the gut may result in the different microbial structures in the midgut and hindgut.

The main objective of this study was to investigate the gut microbial structures, compositions, and function capacities of different gut sections of swamp eel using 16S rRNA gene sequencing. We wondered whether the different structural and functional characteristics of the gut microbial community in different sections were correlated with swamp eel's metabolism of nutrients, immune modulation, and host-microbe interactions. This study would provide the first glimpse of the gut microbiome of the swamp eel.

## METHODS

### Sample collection

Swamp eels (40–45 g) were sampled from a commercial swamp eel farm in Jiangxi Province, China (28.4219 N, 116.4126 E) and were acclimated in dechlorinated tap water at 25 °C in 10 L aquarium tanks. The swamp eel individuals were then fed with minced fish once a day for 8 weeks until dissection. The dechlorinated tap water was changed every day. All experimental swamp eels were healthy and had not received any antibiotics, probiotics, or prebiotics during the feeding period. Fecal samples were collected immediately and separately before euthanasia. Fish were anesthetized with tricaine methanesulfonate and the whole intestines were aseptically removed from the abdominal cavity. The intestine was further dissected using sterile instruments to separate the midgut (immediately after the stomach) and hindgut (immediately before the anus) sections according to *Liem (1967)*. The contents in each gut section were squeezed out and collected separately. The proximal and distal sections of the intestine were then washed with sterile PBS three times to remove remnants of the gut content. The gut mucosa was then scraped off with a sterilized forcep and transferred into a microcentrifuge tube. All samples from different gut sections were used separately for sequencing. All animal procedures were conducted according to the guidelines for the care and use of experimental animals established by the Ministry of Agriculture of China (No. SCXK YU2005-0001). The Animal Care and Use Committee of Jiangxi Agricultural University gave special approval to this study.

### DNA extraction and 16S rRNA gene sequencing

Total DNA was extracted from the gut content and gut mucosa of different individuals using the PowerSoil® DNA Isolation Kit (Mo Bio, San Diego, CA, USA) according to

the manufacturer's instruction. Fecal DNA extraction was performed using the QIAamp Stool Mini Kit (QIAGEN, Hilden, Germany). The barcoded fusion forward primer 338F (5′-ACTCCTACGGGAGGCAGCA-3′) and the reverse primer 806R (5′-GGACTACHVGGGTWTCTAAT-3′) were used to amplify the V3–V4 hyper variable region of the 16S rRNA gene. The Barcoded V3–V4 amplicons were sequenced using the paired-end method on the Illumina MiSeq 2 × 300 platform (Illumina, San Diego, CA, USA) following the standard protocols.

### 16S rRNA gene sequencing data analysis

To obtain clean data, the barcodes and low quality sequences were filtrated using a FASTX-Toolkit. FLASH software was used to merge high-quality paired-end reads into tags (*Magoc & Salzberg, 2011*). Operational Taxonomic Unit (OTU) picking was performed using the USEARCH pipeline with a 97% sequence identity (*Edgar, 2010*). We performed taxonomic assignments for the aligned sequences using the Ribosomal Database Project classifier program with an 80% confidence threshold (*Wang et al., 2007*). Microbial taxa abundance and diversity indices were generated using Quantitative Insights Into Microbial Ecology (QIIME) (*Caporaso et al., 2010*). Phylogenetic investigation of communities by reconstruction of unobserved states (PICRUSt) was used to predict the functional profile of the microbial community (*Langille et al., 2013*). We extracted the closed reference OTU table by comparing quality control reads in QIIME against the Greengenes database. OTU normalization, gene family abundance prediction and function categorization based on Kyoto Encyclopedia of Genes and Genomes (KEGG) pathway was performed using PICRUSt according to the default settings.

### Statistical analysis

Microbial species richness was analyzed using the observed number of OTUs. Principal Coordinate Analysis (PCoA) of beta diversity was performed based on the unweighted and weighted distance matrix. Permutational Multivariate Analysis of Variance was performed to identify section-specific enriched microbial taxa and functional capacities (*Nielsen et al., 2017*). All output results were visualized using ggplot2 and gplots in R package, except the Venn diagrams which were drawn using the online tool (bioinformatics.psb.ugent.be/webtools/Venn/).

## RESULTS

Both data sets are accessible through NCBI's SRA, under study accession number SRP145040.

### Microbial diversities and compositions in different gut sections

Initiallly, 405, 642, 227, 372 and 171 OTUs were identified in the midgut content, midgut mucosa, hindgut content, hindgut mucosa and stools, respectively (Fig. 1A). We then identified specific and common OTUs in different sections via a Venn diagram (Figs. 1B–1D). A total of 63 common OTUs were detected in the midgut content, hindgut content and stools. A total of 315 OTUs were shared by both the midgut mucosa and hindgut mucosa. Notably, we found 54 common OTUs as a core microbiota presented in

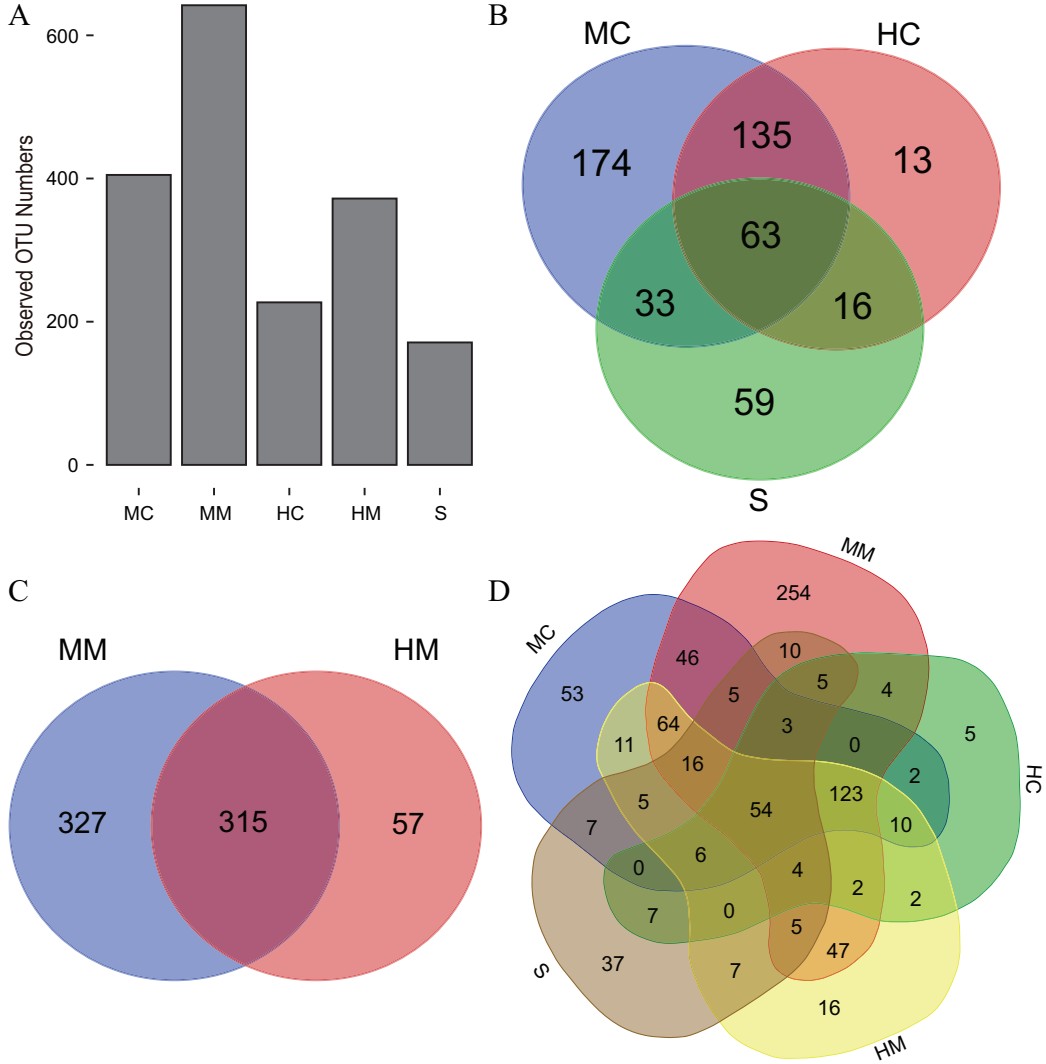

**Figure 1 The observed OTU numbers, unique and shared OTUs in different gut compartments ($n$ = 4).** (A) Bar plot shows the observed OTU numbers in midgut content (MC), midgut mucosa (MM), hindgut content (HC), hindgut mucosa (HM) and stools (S). (B) Venn diagram displays the number of shared and unique OTUs among MC, HC and S. (C) Venn diagram displays the number of shared and unique OTUs between MM and HM. (D) Venn diagram displays the number of core OTUs shared by all gut compartments.

all intestinal sections, while 53, 254, 5, 16, 37 specific OTUs were also detected in midgut content, midgut mucosa, hindgut content, hindgut mucosa and stools, respectively. Moreover, PCoA analysis also revealed significant separations among samples from different gut sections (Fig. 2; Fig. S2).

To further uncover the microbial composition characteristics in different gut sections, we analyzed the OTUs assigned for the phylum and genus levels (Fig. 3). At the phylum level, *Firmicutes*, *Fusobacteria*, *Proteobacteria*, *Bacteroidetes*, and *Actinobacteria* were the five most dominant phyla. At the genus level, *Cetobacterium*, *Ralstonia* and *Rhodococcus* were the most predominant genera. Interestingly, the abundances of these

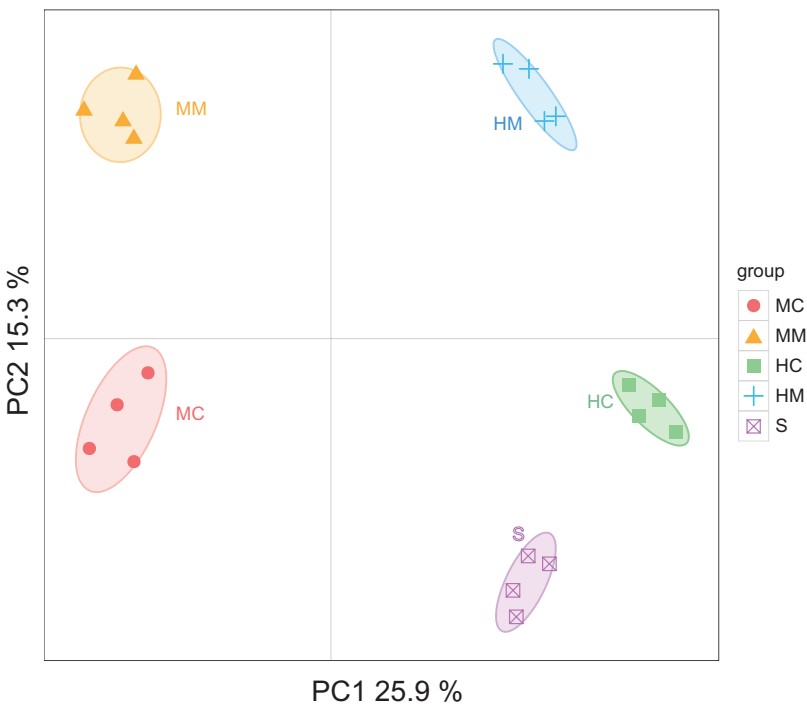

**Figure 2 Principal Coordinate Analysis (PCoA) of the microbial community in different gut compartments based on the Unweighted UniFrac distance matrix ($n$ = 4).** The individual samples are color- and shape-coordinated according to the gut compartment. Midgut content (MC), midgut compartment (MM), hindgut content (HC), hindgut mucosa (HM) and stools (S).

microbial taxa changed significantly across different gut sections. For instance, *Firmicutes* occupied a large proportion of the gut microbiota in both the midgut and hindgut regardless of the location of the sample obtained, but it only occupied a small proportion of the gut microbiota in stool samples. *Fusobacteria* accounted for a higher proportion of gut microbiota in the content section than in the mucosal section. *Cetobacterium* was predominant in all samples, but a lower abundance in the midgut was observed when compared to the hindgut and stools. In contrast, the abundance of *Rhodococcus* in the midgut was higher than that in the hindgut and stools.

## Core microbial taxa enriched in different gut sections

To identify which core microbial taxa showed different enrichment in specific gut sections, we analyzed the abundance of the 54 core OTUs across all sections. Figure 4 shows the total 36 section-specific enriched OTUs. In the midgut content, seven enriched OTUs were annotated to *Enhydrobacter*, *Comamonadaceae*, *Caulobacteraceae*, *Microbacteriaceae*, *Peptostreptococcaceae*, *Bradyrhizobium* and *Deinococcus*, respectively. Meanwhile, OTUs annotated to *Roseburia*, *S24-7*, *Bacillus*, *Acidobacteria*, *Paracoccus*, *Lactococcus* and *Oxalobacteraceae* were enriched in the midgut mucosa. OTUs enriched in the hindgut content were annotated to *Cetobacterium somerae*, *Arthrobacter*, *Coprococcus*, *Bacteroidaceae*, *Ruminococcaceae*, *Epulopiscium* and *Citrobacter*. OTUs annotated to *Clostridium*, *Pseudomonas*, *Rhodococcus*, *Ralstonia*, *Achromobacter*, *Streptococcus* and

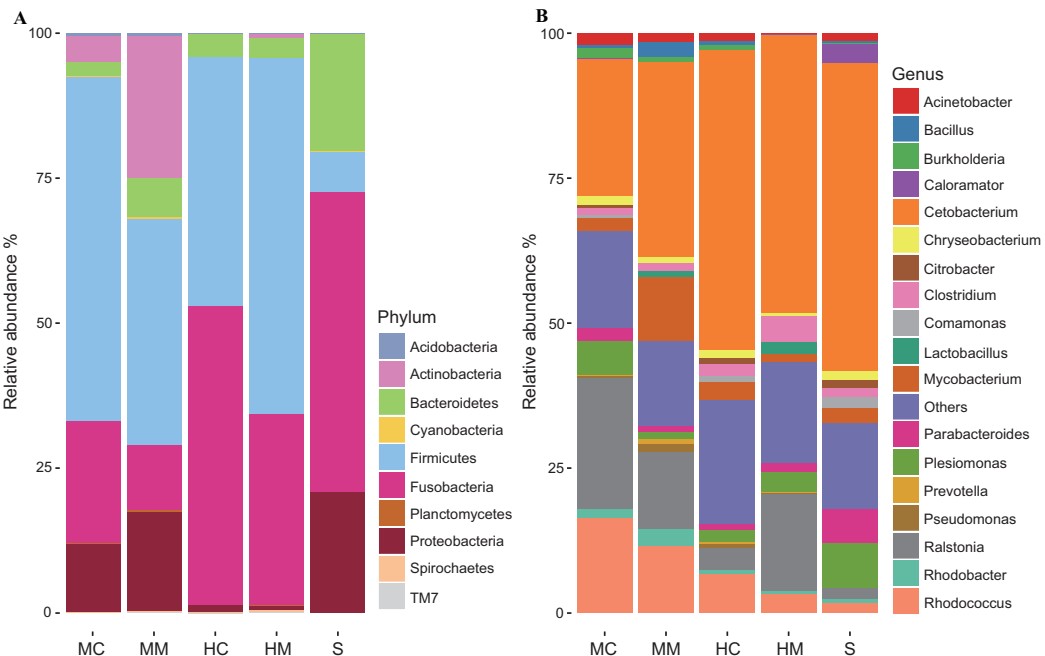

**Figure 3 Gut microbial compositions at the phylum level and genus level (*n* = 4).** (A) Each bar represents average relative abundance of each phylum of gut microbiota in midgut content (MC), midgut mucosa (MM), hindgut content (HC), hindgut mucosa (HM), and stools (S). (B) Each bar represents average relative abundance of each genra of gut microbiota in MC, MM, HC, HM and stools.

*Lactobacillus* showed great abundance in the hindgut mucosa. Finally, eight OTUs derived from *Chryseobacterium*, *Comamonas*, *Serratia*, *Acinetobacter johnsonii*, *Pedobacter*, *Plesiomonas shigelloides*, *Pseudoxanthomonas Mexicana* and *Aeromonadaceae* increased in abundance in the stool samples.

## Comparison of microbial potential capacities in different gut sections

To compare the potential functional capacity of the microbial communities in different gut sections, the relative abundances of KEGG pathways were predicted using PICRUSt. The results showed that 66 KEGG pathways exhibited significant differences in abundances across different gut sections (Fig. 5): 26 pathways from the midgut samples, 28 pathways from the hindgut samples, and 12 pathways from stool samples. Notably, there were some characteristics of the distribution of differential pathways shown in the specific gut section. For example, amino acid metabolism pathways such as lysine degradation, arginine and proline metabolism, and valine, leucine and isoleucine degradation were predominant in the midgut content. Cofactor and vitamin metabolism and signal transduction-related pathways were overrepresented in the midgut mucosa. In the hindgut, carbohydrate and lipid metabolism pathways were prominent in the content, while bacterial replication, transcription, and translation-related pathways were plentiful in the mucosa. Additionally, we observed that microbial communities were more capable of metabolizing secondary metabolites and xenobiotics in the stools.

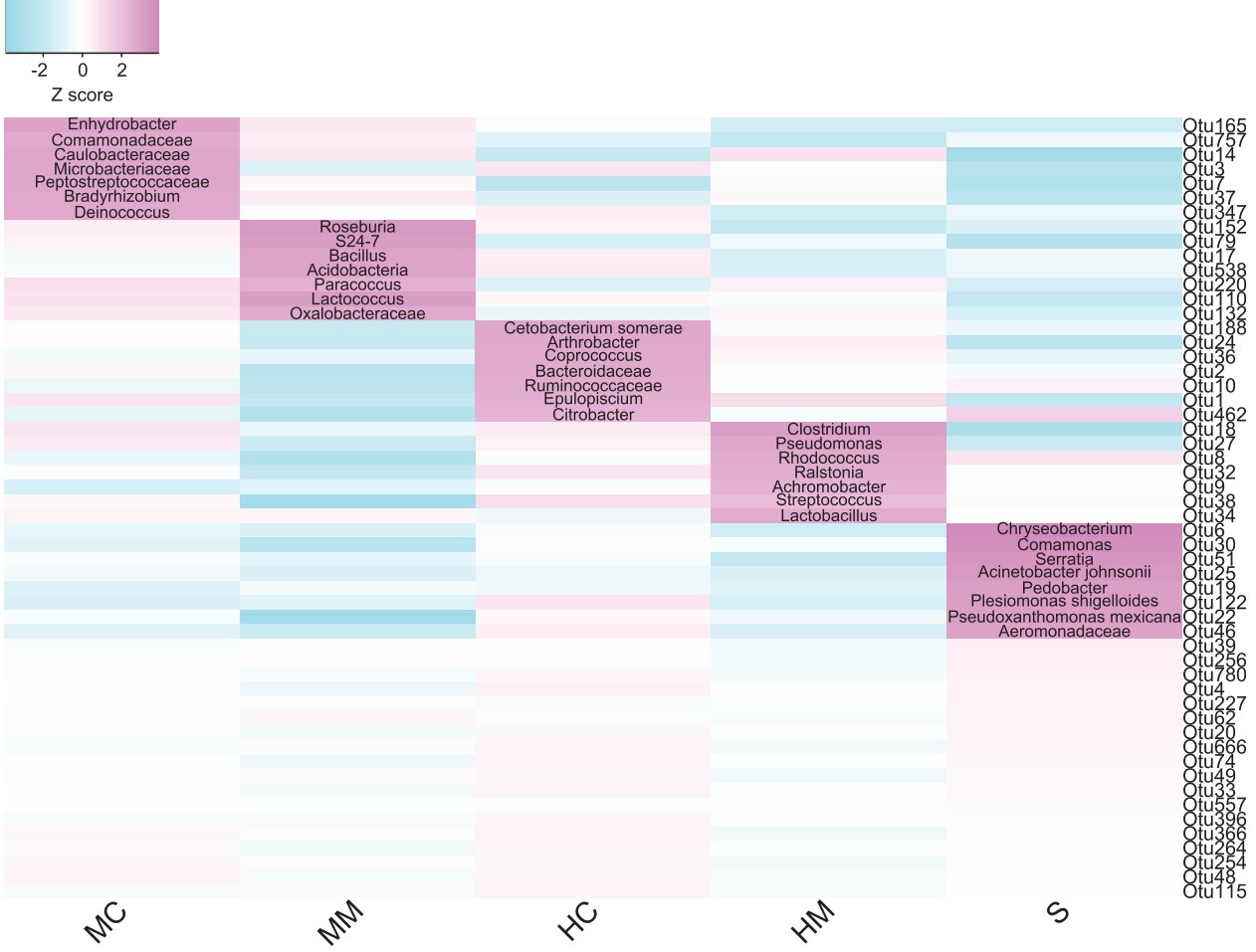

**Figure 4 Gut compartment-specific enriched core OTUs (*n* = 4).** Heat map shows core OTUs significantly varied in abundances in different gut compartments (cell note on the heat map represents differentially abundant OTUs annotated to microbial taxa).

## DISCUSSION

In this study, 16S rRNA sequencing analysis revealed the diversity, composition, and potential functional capacity of microbial communities across different gut sections in swamp eel. To the best of our knowledge, this is the first study systematically evaluating the gut microbiome of swamp eel (*Monopterus albus*).

At the phylum level, *Firmicutes*, *Fusobacteria*, *Proteobacteria*, *Bacteroidetes*, and *Actinobacteria* were the five most dominant phyla. At the genus level, *Cetobacterium*, *Ralstonia*, and *Rhodococcus* were the most predominant genera in the intestinal microbiota communities of swamp eel. Data analysis also showed that the majority of the microbiome found in the intestine of the swamp eel has been detected in other fish, which was consistent with the results found in Japanese eels (*Hsu et al., 2018*). However, when compared with Anguillid eel species, the intestinal microbial composition of swamp eel was markedly different. At the phylum level, *Proteobacteria*, *Fusobacteria*, and

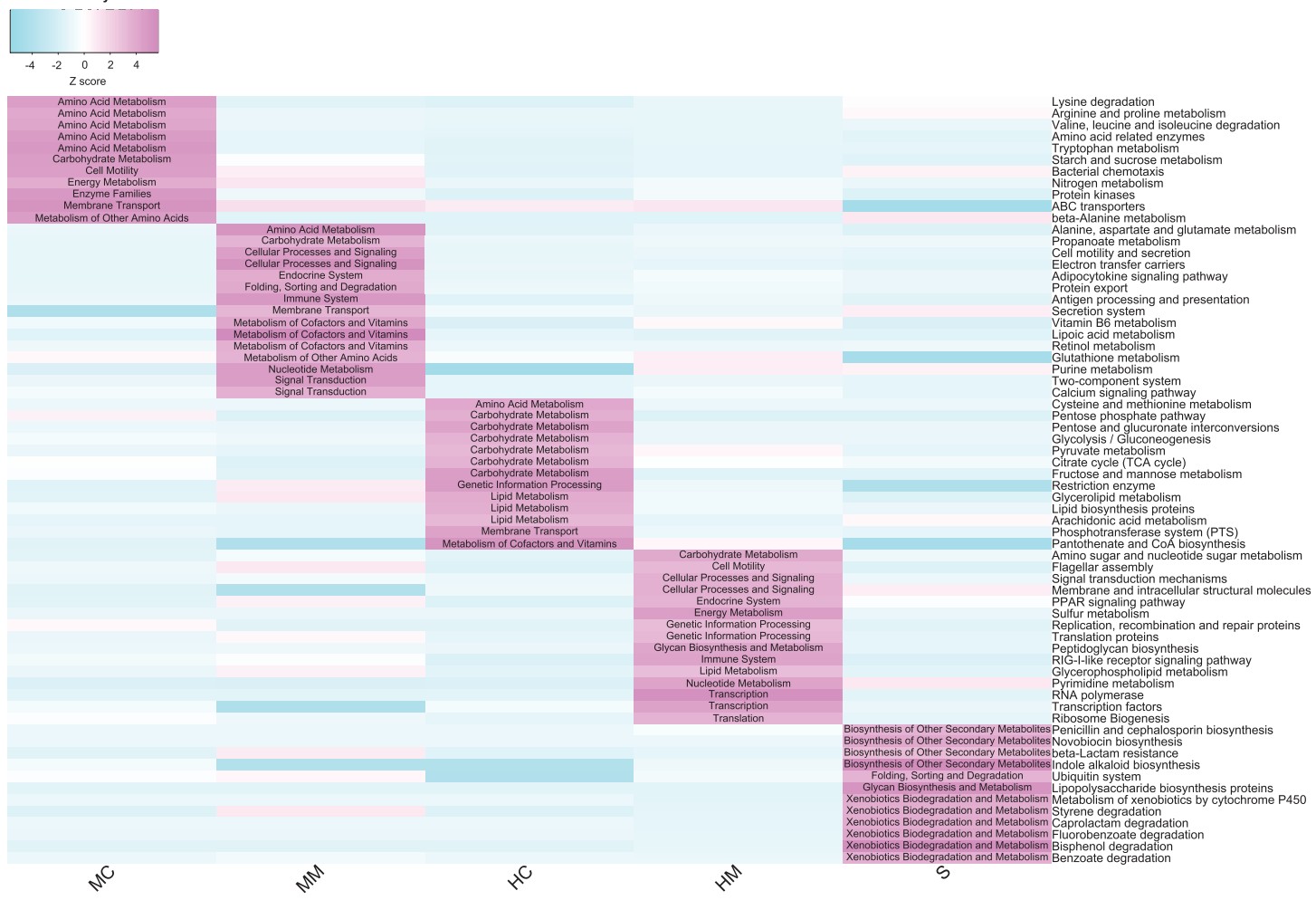

**Figure 5 Comparison of the abundance of gut microbial potential function capacities in different gut compartments (*n* = 4).** Heat map shows the abundances of gut microbial KEGG pathways (level 3) changed significantly in different gut compartments (cell note on the heat map represents differentially abundant KEGG pathways at level 2).

*Bacteroidetes* were the dominant bacterial groups in European eels, and *Proteobacteria* was the most abundant phylum, accounting for 70.35 ± 17.2% of the total number of reads (*Huang et al., 2018*). In swamp eel, *Proteobacteria* accounted for only 12.88% of the total number of reads and were fewer in the hindgut mucosa and hindgut content, accounting for 2.82% and 1.17% of the sequenced reads, respectively. The difference in genus is more obvious. The top five genera in the intestinal mucosa of swamp eel were *Cetobacterium*, *Ralstonia* and *Rhodococcus*, *Mycobacterium*, and *Clostridium*. However, the top five bacterial genera in European eel intestine were *Aeromonas*, *Cetobacterium*, *Plesiomonas*, *Shewanella*, and *Paludibacter*. In Giant-Mottled eels (*Anguilla. marmorata*) the dominant bacterial genera were *Acinetobacter*, *Mycoplasma*, and *Shewanella*. These differences may be related to the genetic characteristics of the species (*Goodrich et al., 2014*; *Li et al., 2018*). Diet is also one of the most important factors that influences community composition (*Moschen, Wieser & Tilg, 2012*; *Piazzon et al., 2017*). In this study, swamp eels
were fed with minced fish, while in other studies, the cultivated European eels were fed with commercial power feed (*Huang et al., 2018*), and the Giant-Mottled eels were caught from the wild (*Hsu et al., 2018*).

In this study, the number of observed OTUs decreased from the proximal to the distal section of the intestine in the swamp eel. This result was different from many vertebrate microbiome studies which showed that the distal section of the intestine had higher richness and diversity than the proximal section. This difference may be associated with the specific physiological structure of the swamp eel's intestine. Unlike omnivorous and herbivorous fish with usually long and coiled intestines (*Pereira et al., 2015*; *Santos et al., 2016*), swamp eel is carnivorous with an intestine that is short and straight. The midgut is thought to be the organ where the majority of digestion occurs (*Egerton et al., 2018*). Considering the crucial role of gut microbiota in host nutrient metabolism, we speculated that the many more microbes inhabiting the proximal section of the swamp eel's intestine increase digestion and absorption of nutrients. Furthermore, there were many more OTUs presented in the mucosal section than in the content section regardless of the locations where samples were obtained. This result was consistent with gut microbiome studies on rabbitfish (*Siganus fuscescens*) and loach (*Nielsen et al., 2017*; *Yang et al., 2017*) and reinforced previous findings that the mucosal section might serve as a reservoir of diverse bacterial species (*Lu et al., 2014*). OTUs assigned for the phylum level and genus level further revealed some specific features of the microbial compositions of swamp eel. A previous study observed a high abundance of *Firmicutes* in the gut microbiome of omnivorous fishes and found that *Fusobacteria* was the predominant phylum in the gut microbiome of carnivorous fishes (*Liu et al., 2016*). Here, the microbial communities of swamp eel were dominated by, in order, *Firmicutes*, *Fusobacteria*, *Proteobacteria*, *Bacteroidetes*, and *Actinobacteria*. Furthermore, we found that *Firmicutes* was more predominant in the midgut and hindgut than in the stools, while the abundance of *Fusobacteria* was higher in the content section than in the mucosal section. The most dominant genera *Cetobacterium*, *Ralstonia*, and *Rhodococcus* also varied in their abundances across different gut sections. These results suggest that using samples from a single gut section to represent an overview of gut microbiota would likely fail to detect community variation responding to physiological variations of the gut (*Durbán et al., 2011*).

Although the gut microbiota showed distinct spatial heterogeneity, we still identified a core microbiota consisting of 54 common OTUs in all gut sections. This was in line with previous fish gut microbiome studies indicating that specific microbial taxa could form a stable core microbota in the intestine (*Baldo et al., 2015*; *Rudi et al., 2018*). Furthermore, we found that the enrichments of these microbial taxa were associated with nutrient metabolism, immune modulation, and habitat adaptions. In the content section, most of the enriched microbial taxa were associated with nutrient metabolism. For example, dietary fiber degradation-associated bacteria *Enhydrobacter* and *Comamonadaceae* (*Premalatha et al., 2015*; *Sakurai et al., 2017*), and amino acid metabolism-associated bacteria *Caulobacteraceae* and *Microbacteriaceae* (*Yin et al., 2017*) were enriched in the midgut content. Gut microbial taxa equipped with multiple carbohydrate active enzymes

such as *Bacteroidaceae*, *Ruminococcaceae*, *Coprococcus*, and *Citrobacter* (*Luo et al., 2017*; *Tap et al., 2015*; *Wu et al., 2012*), which are involved in non-digestible dietary carbohydrate metabolism, showed great abundance in the hindgut content. Notably, *Cetobacterium somerae*, a vitamin B-12 and antimicrobial metabolite-producing species, had a higher abundance in the hindgut content, similar to the results of many other fish gut microbiota studies (*Bledsoe et al., 2016*; *Larsen, Mohammed & Arias, 2014*).

Interestingly, when looking at stool samples as an end product of nutrient metabolism in the content section, we found that several potential aquatic pathogenic bacteria were enriched, including *Serratia*, *Acinetobacter johnsonii*, *P. shigelloides*, and *Aeromonadaceae* (*González et al., 2000*; *Martins et al., 2013*; *Nadirah et al., 2012*; *Huang et al., 2018*). *P. shigelloides* causes diarrhea in humans, usually isolated from the feces (*Khan et al., 2004*), and has also been found in the gut of many fishes, such as tilapia (*Oreochromis niloticus*) (*Nadirah et al., 2012*), largemouth bass *Micropterus salmoides* (*Larsen, Mohammed & Arias, 2014*) and Japanese eel *Anguilla japonica* (*Hsu et al., 2018*). In grass carps, *P. shigelloides* has also been found to be associated with muscle erosive disease (*Hu et al., 2014*). *Acinetobacter johnsonii* were recently shown to be opportunistic pathogens for farmed rainbow trout (*Kozińska et al., 2014*) and blunt snout bream *Megalobrama amblycephala* (*Cao et al., 2017*). However, they do not cause any infections or diseases in our feeding swamp eels. This indicated that these bacteria were native inhabitants of swamp eel stools and the intestine may have a certain ability to enrich harmful bacteria into feces and excrete them out of the body.

Meanwhile, many immune modulation-associated bacteria were found inhabiting the mucosal section. For instance, *S24-7* modulated mucosal immune homeostasis and *Roseburia* regulated innate immunity (*Liu et al., 2017*; *Patterson et al., 2017*). Potential probiotics including *Bacillus*, *Acidobacteria*, and *Lactococcus* (*Bernardeau et al., 2017*; *Lv et al., 2016*; *Wu et al., 2018*) were predominant in the midgut mucosa, and *Lactobacillus* were predominant in the hindgut mucosa. *Clostridium* and *Lactobacillus* involved in immune response, *Pseudomonas* and *Achromobacter* with strong antimicrobial activities, and *Rhodococcus* showing probiotic properties (*Nayak, 2010*; *Sharifuzzaman et al., 2017*; *Zothanpuia et al., 2016*) were overrepresented in the hindgut mucosa. It is noteworthy that aerobic bacteria *Bradyrhizobium*, *Deinococcus*, *Arthrobacter*, and *Comamonas* prefer to thrive in the content section, and anaerobes and obligate anaerobes such as *Paracoccus*, *Ralstonia*, and *Streptococcus* are more prevalent in the mucosal section. These section-specific distributions might be related to special respiration. *Monopterus albus* is an air-breathing teleost using the buccopharyngeal cavity for gas exchange (*Damsgaard et al., 2014*) and this likely causes a small amount of air to enter the intestine. Although there has been no direct study on the oxygen concentration in the intestinal tract of *Monopterus albus*, early literature suggested that the intestine of *Monopterus albus* might have a respiratory function (*Petukat, 1965*).

The potential functional capacities of microbial communities were distinctly different across different gut sections and these differential microbial functional capacities are probably related to host physiological functions and host-microbe interactions. Amino acid metabolism pathways were more abundant in the midgut content, suggesting that gut

microbiota in the midgut content may help swamp eels digest dietary amino acids. Cofactor and vitamin metabolism and cellular signal processing pathways were enriched in the midgut mucosa. Since fishes lack the biosynthetic capacity for most vitamins, it is important that vitamins produced by gut microbiota play a key role in host growth, intestinal mucosal immune, and signaling molecule expression (*Feng et al., 2016*; *Li et al., 2015*). In the hindgut content, a high level of carbohydrate and lipid metabolism was identified. This result was in line with previous studies where gut microbiome of fish hindgut had the ability to ferment non-digestible polysaccharides to short-chain fatty acids (*Geraylou et al., 2013*; *Mountfort, Campbell & Clements, 2002*). In the hindgut mucosa, microbial replications, transcriptions, and translation-related pathways were concentrated, which was consistent with previous studies where hindgut mucosa was an essential gut region where interactions between gut microbiota and host cells occurred (*Morgan et al., 2015*; *Sellers & Morton, 2014*). Intriguingly, we observed that microbial xenobiotics and secondary metabolite metabolism pathways were more predominant in the stool samples. This result combined with the enriched microbial taxa in stools above indicate that stools may serve as a "waste dump" for swamp eels and their microbial community.

## CONCLUSIONS

In the present study, we comprehensively characterized the microbial communities in different gut sections of swamp eel. Our results showed that microbial diversity, composition and function capacity varied substantially in longitudinal and radial sections of the intestine. The microbial diversity and composition across different gut sections could reflect the characteristics of swamp eel's intestinal structures and feeding habits. The gut section-specific enriched core microbial taxa and function capacities may play an important role in swamp eel's nutrient metabolism, immune modulation, and host-microbe interactions. Taken together, these results should provide a basis for further research on the gut microbiome of the swamp eel.

### Funding

This work was supported by grants from the National Natural Science Foundation of China (Nos. 31160530 and 31360634), the Science and Technology Support Program of Jiangxi Province (20122BBF60074) and the Project of Education Department in Jiangxi Province (GJJ13289). The funders had no role in study design, data collection and analysis, decision to publish, or preparation of the manuscript.

### Grant Disclosures

The following grant information was disclosed by the authors:
National Natural Science Foundation of China: 31160530 and 31360634.
Science and Technology Support Program of Jiangxi Province: 20122BBF60074.
Project of Education Department in Jiangxi Province: GJJ13289.

## Competing Interests

The authors declare that they have no competing interests.

## Transparency Document

The Transparency document associated with this article can be found in the attachment.

## Author Contributions

- Xuan Chen conceived and designed the experiments, performed the experiments, prepared figures and/or tables, authored or reviewed drafts of the paper, approved the final draft.
- Shaoming Fang analyzed the data, prepared figures and/or tables, approved the final draft.
- Lili Wei analyzed the data, contributed reagents/materials/analysis tools, prepared figures and/or tables, approved the final draft.
- Qiwang Zhong conceived and designed the experiments, contributed reagents/materials/analysis tools, authored or reviewed drafts of the paper, approved the final draft.

## Animal Ethics

The following information was supplied relating to ethical approvals (i.e., approving body and any reference numbers):

The Animal Care and Use Committee (ACUC) of Jiangxi Agricultural University specially approved this study (JXAUCC-20170015). All experimental procedures were carried out in accordance with China Law for Animal Health Protection and Instructions guidelines (Order No. 2 of the National Science and Technology Commission 1988).

## Data Availability

Both data sets are available at NCBI SRA: SRP145040.

The 16S rRNA gene sequencing data is also available at SRA: SRR7136890 (Bacteroides coprophilus), SRR7136891 (Arthrobacter psychrolactophilus), SRR7136892 (Clostridium perfringens), SRR7136893 (Ruminococcus flavefaciens) and SRR7136894 (Cetobacterium somerae).

## Supplemental Information

Supplemental information for this article can be found online at http://dx.doi.org/10.7717/peerj.8176#supplemental-information.

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
