# Peer review of "Systematic evaluation of the gut microbiome of swamp eel (Monopterus albus) by 16S rRNA gene sequencing"

_PeerJ, doi:10.7717/peerj.8176_

## Round 0.1 · original submission · Major Revisions

· Academic Editor

Major Revisions

Please reply to all the comments point-by-point in your rebuttal letter, with special emphasis in the experimental design of your work including the statistical analysis used.

Reviewer 1 ·

Basic reporting

The authors are recommended to re-check Peer J authors instructions:

1) about the reference guidelines, especially how to include references in the text (e.g. line 54 – 55, 59, 63,65, 69, 124, 196)
and

2) about Style Considerations, "Species formatting" section and correct where is necessary (e.g. line 142)

Experimental design

1) The net cages location of the commercial swamp eel farm should be mentioned, and if/ where and how they have been transferred. Please also provide more information’s about the husbandry techniques and the experimental design.

2) Why choose to use two different isolation kits? Any reason?

3) Is not clear what has been used for the DNA isolation and referred as mucosa in the experiment. Is the gut tissue? Make more clear description for the different sample categories.

4) Please rephrase the sentence line 106 – 107 “The raw sequencing data were removed the barcodes and low quality sequences to obtain the clean data using hASTX-Toolkit.”

Validity of the findings

1) Please rephrase the sentence line 140 – 141 “To further uncover characteristics of microbial compositions in different gut compartments, relative abundances of OTUs assigned for the phylum level and the genus level were analyzed (figure 3).”

2) Line 153 – 155 “To identify the differential enrichment of the core microbial taxa in specific gut compartment, we analyzed the abundance of the 54 core OTUs across all compartments. As shown in figure 4, total 36 compartment-specific enriched OTUs were observed.”

Add all the 54 core OTUs in the heat map to make clear your result.

3) Enrich the discussion. I suggest improving it by discussing findings from relevant studies (e.g. gut microbiota from other eel species)

Additional comments

I recommend the authors to use transition words (moreover, furthermore, in addition) instead of "what's more" (line 67, 138, 179, 210).

I would like also to suggest the authors to use “gut section” instead of “gut compartment”

·

Basic reporting

The paper entitled “Systematic evaluation of the gut microbiome of swamp eel (Monopterus albus) by 16S rRNA gene sequencing” aims at characterizing the microbiota and their predicted functions associated to the foregut, hindgut, and stools of swamp eel using the sequencing of the V3-V4 hypervariable regions of the 16S rRNA gene. It is a manuscript with adequate experimental design and data analysis. There are some problems with the writing. Some sentences are awkward and unclear or not a complete sentence. The manuscript would benefit greatly from having someone with a good command of English proofreading it, particularly on sentence structure and word choices. While the English is adequate for general understanding, it is still difficult to follow the authors' arguments.
Nevertheless, there are few major issues with either the experimental design or data analysis that prevents from completely supporting the claims and conclusions made in the manuscript.

Literature references are sufficient field background/context. The results are relevant results to hypotheses.

Experimental design

Nevertheless, there are few major issues with either the experimental design or data analysis that prevents from completely supporting the claims and conclusions made in the manuscript.

1. The alpha diversity analysis shows a difference between analyzed sections of the swamp eel. It seems that author used the total sequencing depth for this analysis. To compare between samples these metrics should be made at the same sequence deep for all the groups of samples. The sequence depth between the treatments could be different and this result is strongly influenced by sequence depth. The authors should demonstrate and discuss that a similar effect is maintained using alpha diversity metrics calculated at the same sequence depth (using at least 1000 iterations). In this manner, they can obtain and compare metrics such as observed OUT’s at the same sequence depth between the groups of samples and also they can calculate p-values for this metric among the tested groups. Thus, authors eliminate the bias that sequence depth actually has on their results. Please include the rarefaction curves used to calculate the observed OTUs for all the samples as a supplementary figure.

2. It is not clear for this reviewer if the authors used the core of OTUs or the differential OTUS to predict the potential function. Please clarify. A PCoA of weighted UNIFRAC distances should be included and discussed as a supplemental figure to improve their results between fish sections. The authors should describe the number of total reads that were unclassified. The database used to classify the sequencing reads was not reported in materials and methods. The authors used SILVA or GreenGenes?

3. Lines 196-200. This result is probably due to that authors used the observed OTUs using different sequencing depth. Please include a table containing the number of reads for each sample and clarify the sequence deep used for alpha diversity analysis (observed OTUs).

4. It is important to acknowledge PiCrust limitations. The method is based on the annotated functions of known genomes in the database to which the 16S sequences were able to be mapped. Thus, is important to include the NSTI values and consider those values in the discussion in the context of literature using PiCrust such as shrimps or human.

5. How the authors define that an OTU was shared; when it appeared in at least one sample or in all the samples? How the authors define that an OUT was specific; when it was not shared between treatments or when it was enriched independently of the number of samples in which was present?

Lines 122-124. Please clarify where the PERMANOVA was used. The article never discusses the result of this analysis.

Validity of the findings

No comment.

Additional comments

Minor changes

Lines 89-90. Were the samples mixed or separately used for sequencing? This point should be clarified.
Line 98. Was the DNA extracted from the total mucosa?
Lines 104. What is the sequencing platform used for the analysis, MiSeq 2x300, 2x150, 2x100?
LiNES 106-108. Please clarify the Phred score used as a cut-off. Doy you removed N’s from the sequences?
Lines 108-109. Dou you removed the singletons from the OTU’s table?

·

Basic reporting

1. It would be benefic for the non-expert readers if the authors could describe, in the Introduction section, the structure and physiology of the different gut compartments of the swamp eel, as well as the feed habit of this fish in captivity and wild life. In this context, please include a hypothesis regarding the possible microorganisms to be detected.
2. Please provide a better definition and reference for probiotic used in aquaculture (L55-56).
3. I could not access to both data sets, please check the accession number.
4. The abstract is not structured according to PeerJ standards.
5. Please include in all the legend’s figures the number of analyzed individuals. In figure 1 indicates if the results showed averages of these individuals.

Experimental design

Method section:
1. Please provide more information about the diet (ingredient composition) and fish weight.
2. Please explain how the gut mucosa was sampled.
3. DNA extraction was performed from the 4 individuals? Please explain.
4. Describe the quality checking performed with the raw sequences

Validity of the findings

1. How many sequences were obtained per samples?
2. Please provide rarefaction curves of the all the sequenced samples to observe the coverage of the sequencing.
3. The OTUs number, %abundance, and other results corresponded to an average of the 4 individuals?
4. L245-250: provide some evidence and references regarding the oxygen concentration in the eel’s intestine. A deeper analysis including all microorganisms detected is necessary to associate aerobic or anaerobic bacteria to a specific intestinal compartment.
5. Please justify the use of 2 DNA extraction kits, and discuss how the use of different methods can influence the bacterial diversity oberved.

---

## Round 0.2 · accepted · Accept

· Academic Editor

Accept

Although your paper is now accepted, you must provide proof of thorough linguistic checks, at the proof reading stage.

Reviewer 1 ·

Basic reporting

The authors have shown a lot of effort to improve the manuscript. However, there are many grammatical, linguistic and spelling mistakes that should be revised.

Experimental design

With the additional informations, the "Methods" section is now more clear and comprehensible

Validity of the findings

No comment

Additional comments

No comment